# Early evolution of beetles regulated by the end-Permian deforestation

**Xianye Zhao[1,2], Yilun Yu[2,3], Matthew E Clapham[4], Evgeny Yan[5], Jun Chen[1,6], Edmund A Jarzembowski[1,7], Xiangdong Zhao[1,2], Bo Wang[1]\***

[1]State Key Laboratory of Palaeobiology and Stratigraphy, Nanjing Institute of Geology and Palaeontology and Center for Excellence in Life and Paleoenvironment, Chinese Academy of Sciences, Nanjing, China; [2]University of Chinese Academy of Sciences, Beijing, China; [3]Institute of Vertebrate Paleontology and Paleoanthropology, Chinese Academy of Sciences, Beijing, China; [4]Department of Earth and Planetary Sciences, University of California, Santa Cruz, Santa Cruz, United States; [5]Palaeontological Institute, Russian Academy of Sciences, Moscow, Russian Federation; [6]Institute of Geology and Paleontology, Linyi University, Linyi, China; [7]Department of Earth Sciences, Natural History Museum, London, United Kingdom

**Abstract** The end-Permian mass extinction (EPME) led to a severe terrestrial ecosystem collapse. However, the ecological response of insects—the most diverse group of organisms on Earth—to the EPME remains poorly understood. Here, we analyse beetle evolutionary history based on taxonomic diversity, morphological disparity, phylogeny, and ecological shifts from the Early Permian to Middle Triassic, using a comprehensive new dataset. Permian beetles were dominated by xylophagous stem groups with high diversity and disparity, which probably played an underappreciated role in the Permian carbon cycle. Our suite of analyses shows that Permian xylophagous beetles suffered a severe extinction during the EPME largely due to the collapse of forest ecosystems, resulting in an Early Triassic gap of xylophagous beetles. New xylophagous beetles appeared widely in the early Middle Triassic, which is consistent with the restoration of forest ecosystems. Our results highlight the ecological significance of insects in deep-time terrestrial ecosystems.

**\*For correspondence:**
bowang@nigpas.ac.cn

**Competing interest:** The authors declare that no competing interests exist.

## Editor's evaluation

The study proposes a new evolutionary-ecological scenario for Late Paleozoic and early Mesozoic beetles, supported by the summary of all available knowledge about early beetle fossils, including analyses of their taxon and morphological diversity and phylogenetic relationships. The effects of xylophagous beetles during the Paleozoic may have played a fundamental role in global biochemical cycles. The results advance our understanding of the evolutionary success of beetles and the many ways in which large environmental changes may affect biodiversity in general.

## Introduction

The end-Permian mass extinction (EPME; approximately 252 million years ago) was the most severe extinction event in the Phanerozoic (*Benton and Newell, 2014*). The EPME was primarily caused by the eruption of the Siberian flood basalts (*Burgess and Bowring, 2015*; *Fielding et al., 2019*), which generated excessive emissions of thermogenic methane, $CO_2$, and $SO_2$ that cascaded rapid global warming (*Wu et al., 2021*; *Black et al., 2018*), oceanic acidification and anoxia/euxinia (*Schobben et al., 2020*), aridification and other shifts in the hydrological cycle (*Sun et al., 2012*), acid rain (*Black et al., 2014*), wildfires (*Shen et al., 2011*), and ozone destruction (*Benca et al., 2018*). The response

**Figure 1.** Examples of Permian beetles. (**A and B**) Tshekardocoleidae, *Moravocoleus permianus* Kukalová, 1969, photograph and reconstruction. (**C and D**) Permocupedinae, *Permocupes sojanensis Ponomarenko, 1969*, photograph and reconstruction. (**E**) Tshekardocoleidae, *Sylvacoleus richteri* Ponomarenko, 1963, elytra photograph. (**F**) Taldycupedinae, *Taldycupes reticulatus Ponomarenko, 1969*, elytra photograph. Scale bars represent 1 mm.

of terrestrial ecosystems to the EPME is quite heterogeneous, probably due to biotic, topographic, and latitudinal differences (*Fielding et al., 2019*; *Zhao et al., 2020*; *Dal Corso et al., 2020*). Moreover, how terrestrial ecosystems were affected during the EPME is still highly controversial (*Benton and Newell, 2014*; *Gastaldo, 2019*). Terrestrial tetrapods and plants are considered to have been severely affected by the EPME mostly based on diversity and taxonomic composition (*Benton and Newell, 2014*; *Viglietti et al., 2021*); however, such mass extinction was questioned by a more comprehensive dataset of plant macro- and microfossils (*Gastaldo, 2019*; *Nowak et al., 2019*). Similarly, Permian insects are thought to have suffered a significant extinction (*Labandeira and Sepkoski, 1993*; *Béthoux et al., 2005*; *Labandeira, 2005*; *Condamine et al., 2020*; *Condamine et al., 2016*), but this was not supported by other molecular phylogenetic and fossil record analyses (*Ponomarenko, 2016*; *Dmitriev et al., 2018*; *Montagna et al., 2019*; *Schachat et al., 2019*). In addition, the ecological response of insects to the EPME remains poorly understood (*Benton and Newell, 2014*; *Schachat et al., 2021*).

Beetles (Coleoptera) are the most speciose group of extant insects (*Stork, 2018*), with a stratigraphic range dating back to at least the lowest Permian (*Ponomarenko, 2016*; *Kirejtshuk et al., 2013*). They have a rich fossil record since the Permian and display a wide array of lifestyles (*Figure 1*; *Ponomarenko, 1969*; *Ponomarenko, 2003*). Their fossil record thus offers a unique and complementary perspective for studying the ecological response of insects to the EPME. The evolutionary history of Coleoptera has been widely investigated through molecular phylogenetic analyses (*Condamine et al., 2016*; *McKenna et al., 2019*; *Zhang et al., 2018*), morphological phylogenetic analyses (*Beutel et al., 2008*; *Beutel et al., 2019*), and fossil record analyses (*Ponomarenko, 2003*; *Ponomarenko, 2016*; *Smith and Marcot, 2015*). Although a long-term Palaeozoic-Mesozoic turnover of beetle assemblages is supported by almost all analyses, the detailed ecological response to the EPME and its explanatory mechanisms remain unclear. Most of the Permian and Triassic beetles

belong to stem groups (extinct suborders or families; *Figure 1*), and thus they show character combinations and evolutionary history that cannot be inferred or predicted from phylogenetic analysis of modern beetles. In particular, two problems were ignored by previous analyses. First, phylogenetic relationships of some key fossils remain poorly resolved, particularly in their evolutionary relationships to modern taxa. Second, there are two complementary taxonomic systems for Permian and Triassic beetles: one is artificial formal taxa (based on isolated elytra that cannot be definitely classified into any natural group), and the other is the natural taxonomy (commonly based on more complete fossils including bodies and elytra). The formal taxa, like trace fossil taxa, lack comprehensive phylogenetic data (*Ponomarenko, 2004*), and thus they cannot be used unreservedly for biodiversity and phylogenetic analyses, but can be helpful in the morphospace analysis. These issues cloud the temporal resolution of coleopteran biodiversity in deep time and complicate the evolutionary trajectory of beetles but can be overcome through a combination of multiple analytical methods. Therefore, taxonomic diversity, morphological disparity, and ecological shifts are best evaluated jointly to better understand how the EPME has shaped the evolutionary history of beetles.

Here, we compile an updated database of beetles from the Early Permian to Middle Triassic based on the taxonomic revision of fossils (including formal taxa). We analyse the evolution of taxonomic diversity, morphological disparity, and palaeoecological shifts of beetles from the Early Permian to Middle Triassic through phylogenetic and palaeoecological reconstructions and morphospace analyses of fossil material. Our results suggest that xylophagous (feeding on or in wood) beetles probably played a key and underappreciated role in the Permian carbon cycle and that the EPME had a profound ecological influence on beetle evolution. These results provide new insights into the ecological role of insects in deep-time terrestrial ecosystems and the ecological response of insects to deforestation and global warming.

## Results

### Taxonomic diversity

We compiled an updated database of beetles (21 families, 125 genera, and 299 species) from the Early Permian to Middle Triassic based on the taxonomic revision of natural and formal taxa (*Figure 2—source data 1*). Our database contains 18 families, 109 genera, and 220 species of natural taxa. There is a steady increase of families from the Early Permian to Middle Triassic, which is consistent with the result of *Smith and Marcot, 2015*, whose analyses were only conducted at the family level. The diversity of natural taxa displays almost the same trajectory at both species and genus levels (*Figure 2C and E*). The diversity is roughly stable in the Early Permian (Cisuralian), mainly represented by Tshekardocoleidae (*Figure 1*), increases rapidly in the Middle Permian (Guadalupian) and Late Permian (Lopingian), with the rise of the major clades Permocupedidae (Permocupedinae and Taldycupedinae) and Rhombocoleidae. Subsequently, it plunges in the Early Triassic and recovered gradually from the Anisian (early Middle Triassic). In the Ladinian (late Middle Triassic), the diversity clearly exceeds that of the Late Permian (*Figure 2C and E*). From the Middle Triassic, the Permian coleopteran assemblage characterized by Tshekardocoleidae, Permocupedidae, and Rhombocoleidae is completely replaced by a Triassic assemblage dominated by Cupedidae, Phoroschizidae and Triaplidae.

Our database also contains 3 families, 17 genera, and 79 species of formal taxa. A considerable proportion of Permian beetles belong to such taxa (Permosynidae, Schizocoleidae, and Asiocoleidae). These formal taxa mostly belong to stem groups, but some should probably be attributed to the two extant suborders Adephaga and Polyphaga. Both species and genus-level diversities of formal taxa show a gradual increase from the Middle to Late Permian, but decrease distinctly from the Triassic (*Figure 2—figure supplement 1*). The mixed taxa diversity (combining natural and formal taxa) displays the same trajectory to that of natural taxa at both species and genus levels (*Figure 2D and F*).

### Phylogeny

We carried out a phylogenetic analysis based on 93 adult and larval characters across 15 natural taxa representing all natural families of Coleoptera from the Early Permian to Middle Triassic (*Figure 3—source data 1*). Our parsimony analysis result is consistent with a previous analysis (*Beutel et al., 2008*), and confirms that Tshekardocoleidae, Permocupedidae (Permocupedinae and Taldycupedinae), and Rhombocolediae are the stem group of Coleoptera (*Figure 3A*, *Figure 3—figure supplement 1*).

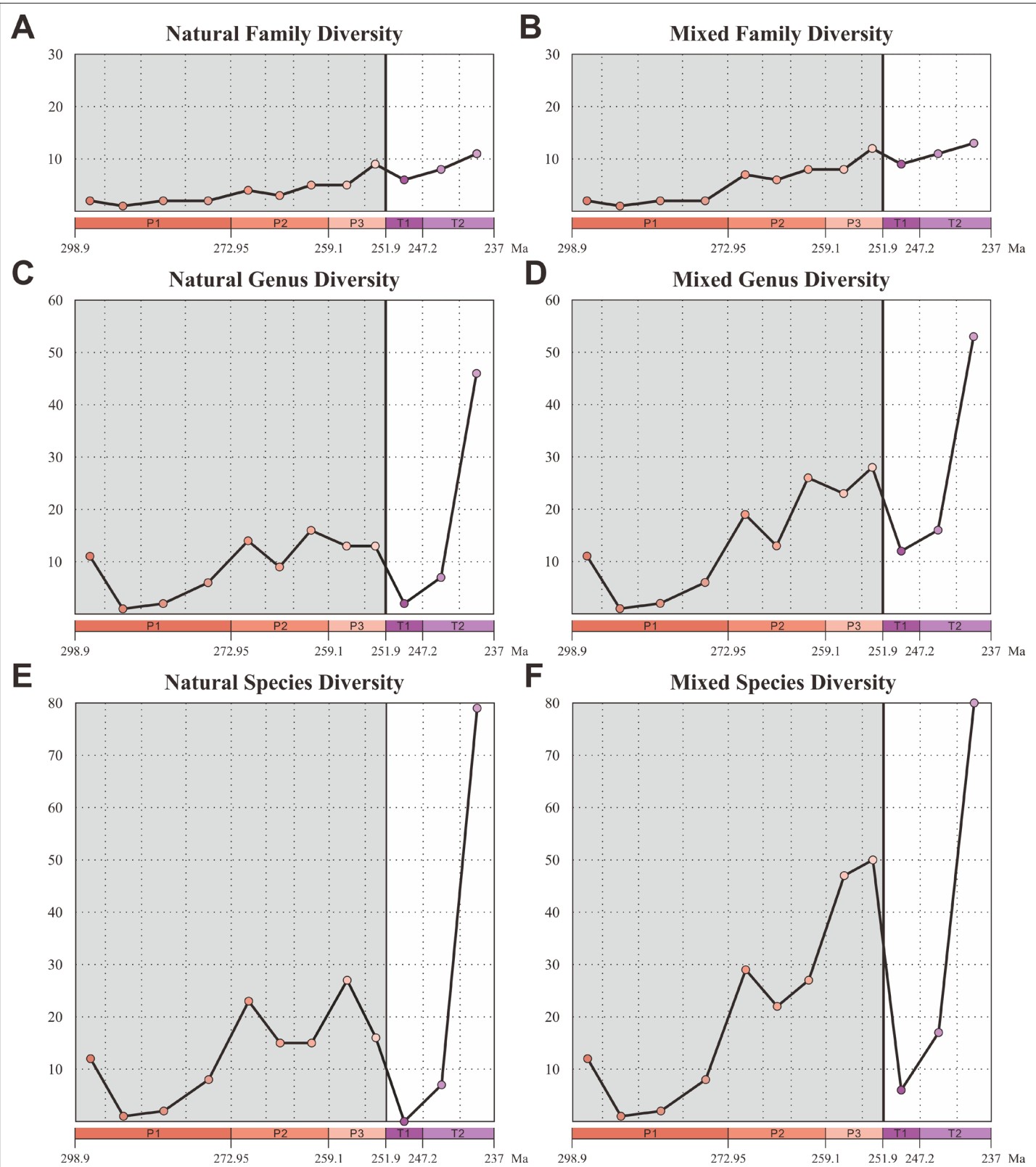

**Figure 2.** Diversity of Coleoptera from the Early Permian to Middle Triassic. Natural taxa and mixed taxa (natural taxa and formal taxa) are counted at family, genus, and species levels separately. (**A**) Family-level diversity of natural taxa. (**B**) Family-level diversity of mixed taxa. (**C**) Genus-level diversity of natural taxa. (**D**) Genus-level diversity of mixed taxa. (**E**) Species-level diversity of natural taxa. (**F**) Species-level diversity of mixed taxa. Abbreviations: P1, Early Permian; P2, Middle Permian; P3, Late Permian; T1, Early Triassic; T2, Middle Triassic.

*Figure 2 continued on next page*

*Figure 2 continued*

The online version of this article includes the following figure supplement(s) for figure 2:

**Source data 1.** Fossil coleoptera database.

**Figure supplement 1.** Diversity of Coleoptera formal groups from the Early Permian to Middle Triassic.

## Morphological disparity

We chose beetle elytra—hardened forewings primarily serving as protective covers for the hindwings and body underneath—to perform the morphological disparity analysis for three reasons: (1) elytra are the most commonly preserved fossils of Palaeozoic and Mesozoic beetles, and they are easily accessible in the literature and in online databases; (2) Permian and Triassic elytra display complex morphological structure (*Ponomarenko, 1969*); (3) elytra morphology has long been studied in relation to taxonomic diversity of living and extinct beetles (*Ponomarenko, 2004*; *Tong et al., 2021*).

We assembled two discrete character matrices (at species and genus levels) based on 35 characters of 197 genera and 346 species (including undetermined species and unnamed specimens) for morphological disparity analyses (*Figure 4—source data 1*). The taxa were ordinated into a multivariate morphospace using both principal coordinates analysis (PcoA) and non-metric multidimensional scaling (NMDS) with two distance metrics, including the generalized Euclidean distance (GED) and maximum observable rescale distance (MORD). We chose both sum of variance (sov) and product of variance (pov) as the proxy for morphological disparity due to their robustness in sample size (*Simões et al., 2020*). The use of discrete characters produces results that have non-metric properties, but this approach can be used to elucidate broad patterns of similarities and clustering within multidimensional space (*Lloyd, 2016*; *Deline et al., 2018*).

The patterns of morphospace occupation of beetles in different time-bins are shown in three-dimensional plots delimited by combinations of the first three axes of the PcoA and NMDS results

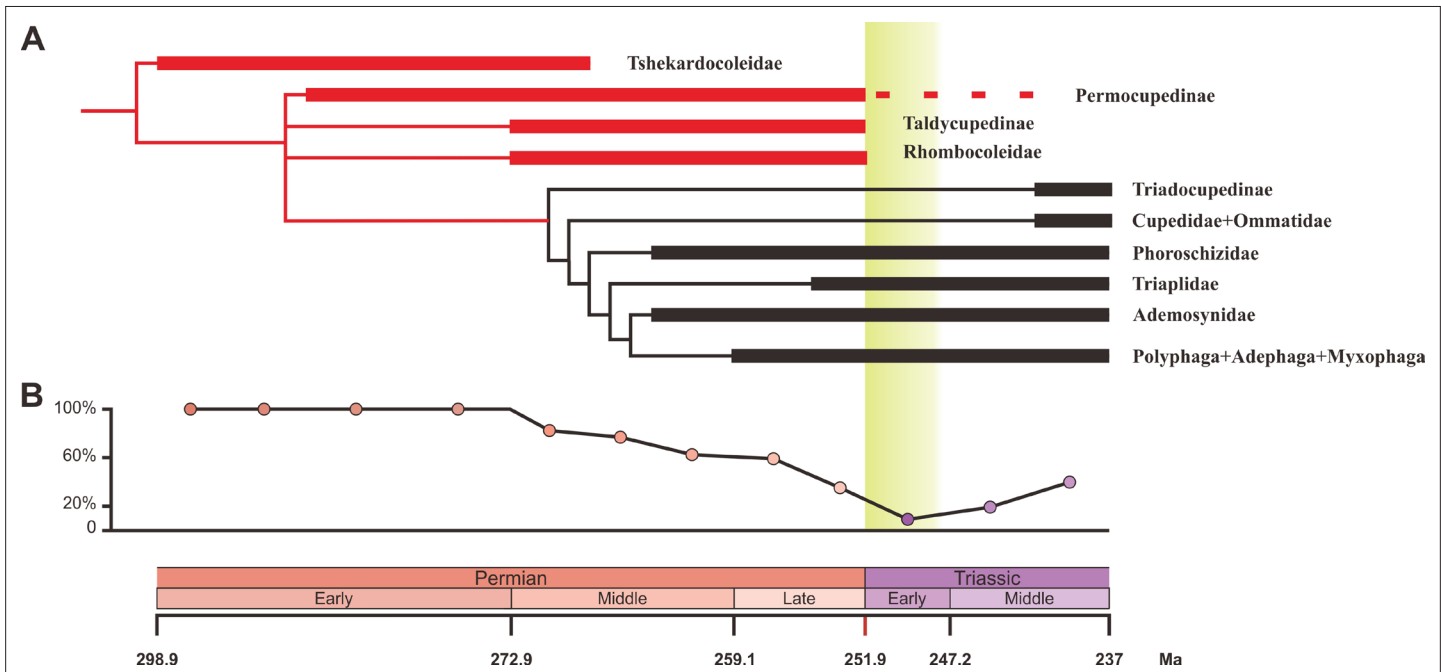

**Figure 3.** Ecological shifts of Coleoptera from the Early Permian to Middle Triassic. (**A**) Simplified phylogeny of Coleoptera from the Early Permian to Middle Triassic. Thick lines indicate the known extent of the fossil record. The branches representing stem groups are shown in red. The 'dead clade walking' pattern is symbolized by the dashed line. For details of the phylogenetic analysis, see *Figure 3—figure supplement 1*. (**B**) Genus percentage of xylophagous groups from the Early Permian to Middle Triassic. Yellow graded band represents the 'coal gap'.

The online version of this article includes the following figure supplement(s) for figure 3:

**Source data 1.** Character state matrix for the phylogenetic analysis.

**Figure supplement 1.** Strict consensus tree of three most parsimonious trees of Coleoptera.

based on MORD metrics (*Figure 4A*, *Figure 4—figure supplement 1*). The morphological disparity results of two ordination methods within the MORD and GED matrix shows the same trajectory at both genus and species levels. The evolutionary pattern of morphological disparity is robust in different disparity metrics. Disparity is low in the Early Permian, with a significant increase during the Middle Permian. It is roughly stable in the Middle and Late Permian, subsequently showing a distinct plunge in the Early Triassic, slightly recovering in the Middle Triassic but is still significantly lower than in the Middle and Late Permian (*Figure 4*, *Figure 4—figure supplements 1–8*).

## Discussion

Our results demonstrate that beetles display a steady accumulation of taxonomic diversity throughout the Permian (*Figure 2*). The earliest definite beetles are Tshekardocoleidae (including the genus *Coleopsis*) from the Early Permian of Germany, Czech Republic, USA, and Russia (*Figure 1*), although the origin of Coleoptera is dated to the Carboniferous by molecular phylogenetic analysis (*McKenna et al., 2019*; *Zhang et al., 2018*; *Toussaint et al., 2016*). The coleopteran diversity radiation in the Middle Permian is consistent with an expansion of morphological disparity, corresponding to the appearance of multiveined, smooth, and striate elytra as well as some other patterns (*Figure 4*, *Figure 4—figure supplement 1*). Taxonomic diversity and morphological disparity were decoupled during the Late Permian when taxonomic diversity increased but morphological disparity was almost stable (*Figures 2 and 4*). The abrupt Middle Permian increase of coleopteran morphological disparity conforms to the early burst model of clade disparity commonly arising early in radiations (*Simões et al., 2020*; *Hughes et al., 2013*).

The Permian coleopteran assemblage was dominated by stem groups including Tshekardocoleidae, Permocupedidae (Permocupedinae and Taldycupedinae), and Rhombocoledidae in terms of richness and abundance. These ancient beetles were most likely xylophagous because they display a prognathous head, a characteristic elytral pattern with window punctures, a cuticular surface with tubercles (or scales), and a plesiomorphic pattern of ventral sclerites, very similar to the extant wood associated archostematans (*Figure 1*; *Kirejtshuk et al., 2013*; *Ponomarenko, 1969*). Moreover, Permian trace fossils showing wood boring provide convincing evidence for the xylophagous habit of these ancient beetles (*Naugolnykh and Ponomarenko, 2010*; *Feng et al., 2019*). Aquatic or semi-aquatic beetles including Phoroschizidae and Ademosynidae, belonging to the suborder Archostemata, first appeared in the Middle Permian and diversified in the Late Permian (*Ponomarenko, 2003*). The three other suborders of Coleoptera, comprising Polyphaga, Adephaga, and Myxophaga, most likely evolved by the Late Permian, but definite fossils are rare at this time.

Permian beetles probably played an important ecological role in forest ecosystems because most Permian beetles were most likely xylophagous insects that consumed living and dead woody stems (*Figure 3*). Some Permian xylophagous beetles fed on living wood tissues (*Feng et al., 2017*; *Feng et al., 2019*), which likely reduced tree productivity and could have caused extensive tree mortality. Insect-mediated tree mortality is known to result in large transfers of carbon from biomass to dead organic matter (*Seidl et al., 2018*; *Fei et al., 2019*). The other Permian xylophagous beetles were likely saproxylic (feeding on dead wood) (*Ponomarenko, 2003*), and they could also impact terrestrial carbon dynamics by accelerating wood decomposition (*Ulyshen, 2018*). Saproxylic animals first appeared in the Devonian and are mainly represented by small invertebrates such as oribatid mites, until the Permian (*Labandeira et al., 1997*; *Labandeira, 2007*). Whereas grazing by micro- and meso-invertebrates (nematodes, collembolans, enchytraeids and oribatid mites) did not significantly affect wood decomposition, consumption by macro-invertebrates (dominated by saproxylic beetles and termites in modern ecosystems) significantly sped up wood decomposition (*Tapanila and Roberts, 2012*). In addition to those that directly facilitated decomposition by consuming wood, Permian saproxylic beetles are likely to have had a variety of indirect effects on decomposition, including creating tunnels that facilitate the movement of fungi into wood (*Naugolnykh and Ponomarenko, 2010*; *Feng et al., 2017*), and vectoring fungi and other decay organisms on or within their bodies, like their extant counterparts (*Ulyshen, 2016*). In conclusion, Permian beetles that feed on living and dead wood probably could impact terrestrial carbon dynamics by reducing forests' carbon sequestration capacity, and by converting live materials to dead organic matter and subsequent decomposition.

The oxygen concentration of the atmosphere began to rise in the early Palaeozoic, probably with a peak in the Carboniferous and large decline from the beginning of the Permian (*Dahl et al.,*

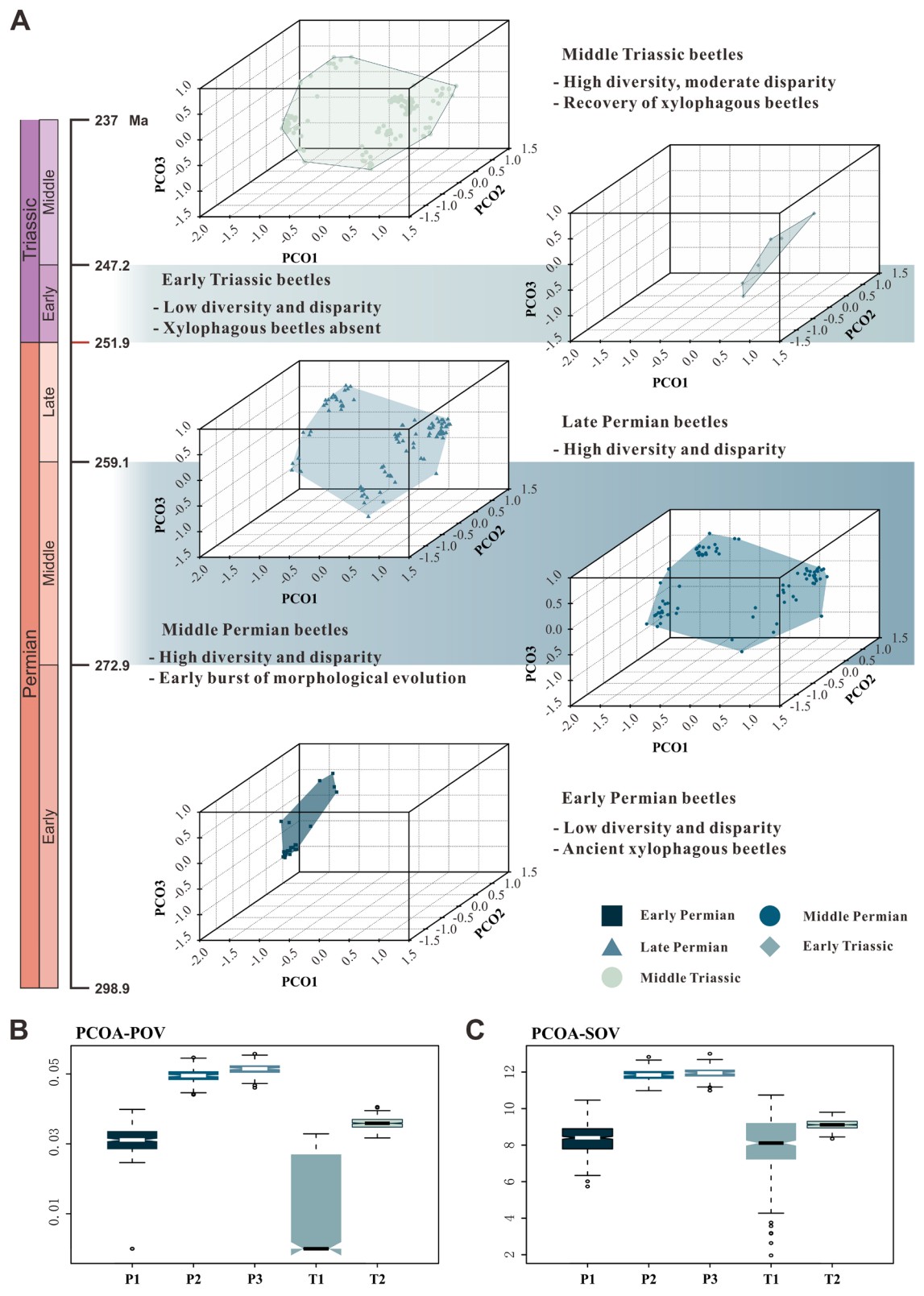

**Figure 4.** Morphospace comparisons of Coleoptera from the Early Permian to Middle Triassic. (**A**) Morphospace three-dimensional (3D) plot ordinated by principal coordinates analysis (PcoA), maximum observable rescale distance (MORD) matrices, based on species-level dataset. (**B and C**) Disparity comparisons ordinated by PcoA, MORD matrices, based on species-level dataset, proxy by pov and sov. Abbreviations: pov, product of variance; sov, sum of variance; P1, Early Permian; P2, Middle Permian; P3, Late Permian; T1, Early Triassic; T2, Middle Triassic.

*Figure 4 continued on next page*

*Figure 4 continued*

The online version of this article includes the following figure supplement(s) for figure 4:

**Source data 1.** Fossil character matrix for the morphospace analysis.

**Source data 2.** Result of permutation test for morphological disparity.

**Figure supplement 1.** Morphospace comparisons of Coleoptera from the Early Permian to Middle Triassic, maximum observable rescale distance (MORD) matrices, proxy by pov and sov.

**Figure supplement 2.** Morphospace comparisons of Coleoptera from the Early Permian to Middle Triassic, based on genus-level disparity analyses of generalized Euclidean distance (GED) matrices, proxy by pov and sov.

**Figure supplement 3.** Morphospace comparisons of Coleoptera from the Early Permian to Middle Triassic, based on species-level disparity analyses of generalized Euclidean distance (GED) matrices, proxy by pov and sov.

**Figure supplement 4.** Permutation tests with sample size corrected (maximum observable rescale distance [MORD]).

**Figure supplement 5.** Permutation tests with sample size corrected (generalized Euclidean distance [GED]).

**Figure supplement 6.** Disparity comparison and permutation tests with sample size corrected, under ordination method of principal coordinates analysis (PcoA), assuming that the age of the Grès à Voltzia specimens is Early Triassic.

**Figure supplement 7.** Disparity comparison and permutation tests with sample size corrected, under ordination method of non-metric multidimensional scaling (NMDS), assuming that the age of the Grès à Voltzia specimens is Early Triassic.

**Figure supplement 8.** Shepard 'goodness-of-fit' stress plot.

---

*2010*; *Berner, 2009*; *Krause et al., 2018*). The reason for this plunge was attributed to a tectonic- or climate-driven reduction in the extent of coal swamps (*Berner and Canfield, 1989*) or to the evolution of lignin-consuming fungi (*Floudas et al., 2012*). However, global recoverable coal is only equivalent to a few percent of the oxygen budget in the atmosphere, and thus cannot account for the large drop of atmospheric oxygen (*Nelsen et al., 2016*). Furthermore, lignin-consuming fungi may have been present before the Carboniferous (*Nelsen et al., 2016*). Recently, a new geochemical model proposed that the development of Permian terrestrial herbivores may have limited transport and long-term burial of terrestrial organic compounds in marine sediments, resulting in less organic carbon burial and attendant declines in atmospheric oxygen (*Laakso et al., 2020*). Herbivorous insects and amniotes are thought to be the major herbivorous animals during the Permian and thus are considered to be the most important drivers of the Permian change in biogeochemical cycles of carbon (*Laakso et al., 2020*). However, Permian herbivorous amniotes mainly fed on leaves, stems, roots, and rhizomes (*Sues and Reisz, 1998*; *Pearson et al., 2013*) and could normally digest cellulose by fermentation but could not consume lignin, as in extant herbivorous vertebrates (*Pearson et al., 2013*). The majority of terrestrial plant biomass is stored in forest woody tissue consisting of decay-resistant lignin (*Hibbett et al., 2016*; *Bar-On et al., 2018*). In modern forests, the total carbon stock in woody tissue (including living and dead wood) is approximately 340 Pg carbon, much more than 72 Pg carbon in roots (below ground), 43 Pg carbon in foliage, and 43 Pg carbon in litter (*Reich et al., 2014*; *Pan et al., 2011*). In extant forest ecosystems, insects may account for 29% of the total carbon flux from dead wood and thus they have a functional importance in the decomposition of dead wood and the carbon cycle (*Seibold et al., 2021*). During the Permian, beetles were probably the dominant consumers of woody tissue, while a few other insect groups may have sometimes fed on dead wood (such as stem dictyopterans and protelytropterans) (*Grimaldi and Engel, 2005*). Permian beetles had probably evolved close interactions with various microorganisms, especially lignin-consuming fungi (*Nelsen et al., 2016*), which also accelerated the decomposition of dead wood. The Early Permian onset of the decrease in oxygen concentrations is consistent with the origin and radiation of the xylophagous beetles in the fossil record. Therefore, we propose that Permian xylophagous beetles could have been responsible for at least part of the change in Permian biogeochemical cycles in Laakso's model (*Laakso et al., 2020*).

As the most taxonomically and functionally diverse group of living organisms on Earth (*Stork, 2018*), extant insects have significant effects on terrestrial carbon and nutrient cycling by modulating the quality and quantity of resources that enter the detrital food web (*Belovsky and Slade, 2000*; *Kurz et al., 2008*; *Yang and Gratton, 2014*; *Seibold et al., 2021*). However, the effects of insects on terrestrial ecosystems in deep time have been viewed as unimportant or overlooked (*Doughty, 2017*). Permian beetles were among the principal degraders of wood and played a fundamental role

in deep-time carbon and nutrient cycling and niche creation. Insects may have been one of the major regulating factors of forest ecosystems at least from the Permian.

Our results show that both the taxonomic diversity and morphological disparity dropped dramatically during the Early Triassic (*Figures 2 and 4*). Combined with the phylogenetic results (*Figure 3*), our suite of analyses yields a clear ecological signal from beetles across the Permian/Triassic boundary: all xylophagous stem-group beetles become extinct near the Permian-Triassic boundary or abruptly decreased in the Early Triassic (a pattern called 'dead clade walking'; *Barnes et al., 2021*), while aquatic phoroschizid and ademosynid lineages crossed the Permian/Triassic boundary and diversified in the Middle Triassic. Coleoptera recovered in taxonomic diversity during the Middle Triassic by the rise of new predatory and herbivorous groups, synchronized with the recovery of terrestrial ecosystems (*Zhao et al., 2020*). However, the morphological disparity is significantly lower than that of the Middle and Late Permian due to the lack of stem-group beetles that possess complex elytra structures (*Figure 4*, *Figure 4—figure supplement 1*). Polyphagan groups increased in taxonomic diversity during the Middle Triassic, which is a transitory epoch from a Palaeozoic stem-group beetle assemblage to a Mesozoic polyphagan-dominated assemblage.

Xylophagous groups are absent or rare in Early Triassic coleopteran assemblages, becoming widespread again from the Middle Triassic, mainly represented by more derived archostematans (such as Cupedidae) and polyphagans (*Ponomarenko, 2003*). This gap in xylophagous beetles coincided chronologically with the gap in coal deposition ('coal gap'), a time during which peat-forming forests were rare or absent (*Figure 3B*), extending across at least the entire Early Triassic (*Benton and Newell, 2014*; *Retallack et al., 1996*; *Nowak et al., 2020*; *Zhao et al., 2020*). During the latest Permian and earliest Triassic, gymnosperm-dominated forests abruptly collapsed (*Vajda et al., 2020*) and were replaced by other biomes (such as isoetalean-dominated herbaceous heathlands; *Feng et al., 2020*) in most areas due to extreme conditions including aridity (*Sun et al., 2012*), wildfires (*Shen et al., 2011*), and ozone destruction (*Benca et al., 2018*). In some regions, the plant extinction was less severe, or the recovery was rapid (*Hochuli et al., 2010*), or there may have been multiple crises during the Early Triassic (*Schneebeli-Hermann et al., 2017*), but even short-term ecosystem disruption could have led to extinctions among xylophagous beetles. Previous studies have not provided a clear picture of insect evolution in response to possible environmental stresses, nor of their response to the EPME (*Benton and Newell, 2014*). Our results show for the first time that the demise of most forests (deforestation event; *Vajda et al., 2020*) most likely resulted in the extinction of most Palaeozoic xylophagous beetles, analogous to the extinction of tree-dwelling birds and mammals resulting from end-Cretaceous deforestation (*Field et al., 2018*; *Hughes et al., 2021*).

Our results reveal an Early Triassic gap in xylophagous beetles, suggesting that early archaic beetles experienced the severe ecological consequences of end-Permian deforestation. Extant insects are suffering from dramatic declines in abundance and diversity largely due to the anthropogenic deforestation and global warming (*van Klink et al., 2020*; *Wagner et al., 2021*). However, xylophagous insects have been largely neglected in studies of the current extinction crisis (*van Klink et al., 2020*). In particular, the diversity and abundance of xylophagous beetles are extremely sensitive to climate change and can also entail forest collapse and carbon cycle disturbance (*Kurz et al., 2008*; *Fei et al., 2019*; *Šamonil et al., 2020*). Our findings may help to better understand future changes in insect diversity and abundance and its consequences faced with global environmental change.

## Materials and methods
### Diversity analysis
We compiled an updated database of all coleopteran species from the Early Permian to Middle Triassic from published literature in the early 1800s through to early 2020. In addition, we incorporated data from other open access database projects, including the Fossil Insect Database (EDNA) and Paleobiology Database (PBDB). We re-examined all published occurrences and taxonomy of Coleoptera from the Early Permian to Middle Triassic (*Figure 2—source data 1*). We standardized and corrected for nomenclatural consistency of all taxa using a classification of extinct beetle taxa above the genus rank (*Bouchard et al., 2011*). The data were filtered and cleaned by removing or reassigning illegitimate, questionable, and synonymous taxa and converting local to global chronostratigraphic units (*Supplementary file 1*).

We allocated fossil species into 12 stage-level time-bins covering the Early Permian-Middle Triassic interval (from the Asselian to Ladinian, 298–237 Ma). Considering the short duration of the Induan and Olenekian stages, we combined both stages into one time-bin. The formal taxa were erected based only on isolated elytra that cannot be classified definitely into any natural group. Thus, we separately counted the diversity of natural, formal, and mixed groups (*Figure 2*, *Figure 2—figure supplement 1*). The species *Coleopsis archaica* was attributed to Tschekardocoleidae by *Kirejtshuk et al., 2013*, but was later elevated to a new family Coleopseidae by *Kirejtshuk, 2020*. We followed the former opinion because it is premature to erect a family without a detailed cladistic analysis. We determined the stratigraphical ranges of families, genera, and species as the maximum and minimum ages in stage-level time-bins. All diversity was calculated using the range-through method.

## Phylogenetic analysis

In light of the new taxa and characters available for further testing the phylogenetic status of ancient stem-group beetles, we reconstructed the phylogenetic relationships among the stem groups by incorporating the presently described new taxa and revised characters coding into the previous dataset (*Beutel et al., 2008*). The morphological characters used for phylogenetic analysis comprise 93 adult and larval characters (*Figure 3—source data 1*). Unknown characters were coded as '?'. The taxon sampling contains two megalopterans as outgroups (*Sialis* and *Chauliodes*) and 13 coleopteran ingroup taxa (five extant and eight extinct) representing all four coleopteran extant suborders and their stem groups (*Supplementary file 2*). Compared to previous character matrices, we added the subfamily Taldycupedinae and three new characters (*Supplementary file 3*). The matrix was analysed in TNT version 1.1, through parsimony analysis and using traditional search (*Goloboff et al., 2008*). All characters were equally weighed and unordered (1000 replicates and 1000 trees saved per replication). Bootstrap values, consistency index, and retention index were provided (*Figure 3—figure supplement 1*).

## Morphospace analysis

We performed morphospace analyses with our newly assembled discrete character matrices (*Figure 4—source data 1*; *Supplementary file 4*). The analyses were performed using the free software R. 4.0.4. Both the MORD matrix and GED matrix were calculated based on two discrete character matrices (*Lloyd, 2016*; *Wills, 1998*). Recent research has revealed that the GED matrix creates a systematic bias in cases with a high percentage of missing data, and the MORD matrix can provide greater fidelity under these circumstances (*Lloyd, 2016*; *Lehmann et al., 2019*). We then ordinated all taxa into a multivariate morphospace with both PcoA and NMDS.

For PcoA, we used the function 'ordinate cladistic matrix' in package 'Claddis' with a cailliez method to correct the negative eigenvalues (*Lloyd, 2016*). Two disparity matrices were used to evaluate the volume of the morphospace, including the sum and product of the variances (*Wills et al., 2016*). The product metrics was normalized by taking the nth root (n equals the number of axes used for calculating disparity metrics). We used the scores on all axes that together comprise 90 % of total variance to calculate those disparity metrics. We chose a permutation test (two-tailed) to test the null hypothesis of no difference between insect disparity of different time-bins. Each test run used 5000 replications. The test statistic was obtained by using the disparity metric of an older time-bin to minus that of a younger time-bin. If the proportion in the null distribution greater than the observed value of the test statistic is smaller than 0.025, the insect disparity of an older time-bin was considered significantly larger than that of a younger time-bin, and if the proportion was greater than 0.975, the insect disparity of a younger time-bin was considered significantly larger than that of an older time-bin (*Figure 4—source data 2*).

We also performed permutation tests with sample size corrected. For two design groups with different sample sizes, we first performed subsampling of the group with more samples to obtain equal sample sizes. Based on the newly obtained two groups with equal sample sizes, we calculated the observed value of the test statistic. Then we randomly permutated those species into different groups once and calculated a test statistic. We repeated this procedure (subsampling and permutation) 10,000 times and obtained a null distribution plus a group of observed values. We then calculated a set of proportions greater than the observed values in the null distribution. By analogy, if the median of the proportions is equal to or smaller than 0.025, the insect disparity of an older time-bin is

significantly larger than that of a younger time-bin. If the median proportion is considered significantly greater than 0.975, the insect disparity of a younger time-bin is larger than that of an older time-bin (*Figure 4—figure supplements 4–7*).

For NMDS, we used the function 'metaMDS' in package 'vegan' with the number of dimension settings to 3 (*Dixon, 2003*). Both non-metric fit and linear fit were very high (larger than 0.90; *Figure 4—figure supplements 4–7*) and the stresses were smaller than 0.2, which implies that the ordinations are relatively good. Then we repeated all the previous analyses with this NMDS morphospace and acquired a similar result. The two disparity metrics of each time-bin were calculated based on two different distance matrices and two different ordination methods (*Figure 4*, *Figure 4—figure supplements 1–5*). The distribution was simulated under 500 bootstraps. Thirty-one undetermined specimens from the Grès à Voltzia Formation (Lower/Middle Triassic boundary, France) were included in our database and their age was attributed to the early Middle Triassic in our analysis (*Figure 4*, *Figure 4—figure supplements 1–5*). Considering that the age of these specimens is controversial, we repeated all our analyses assuming that the age of these specimens is Early Triassic; the result is consistent with the previous one (*Figure 4—figure supplements 6 and 7*).

## Acknowledgements

We are grateful to George Perry, Martin Fikáček, and two anonymous reviewers for invaluable comments that improved this manuscript. We thank AG Ponomarenko, RG Beutel, AP Rasnitsyn, AG Kirejtshuk, J Xue, H Xu, B Huang, and H Zeng for helpful discussion, and D Yang for reconstructions. BW thanks members of the palaeoentomological laboratory of the Palaeontological Institute (Russian Academy of Sciences) for their help during his visit to Moscow (2010).

## Additional information

### Funding

| Funder | Grant reference number | Author |
| --- | --- | --- |
| Chinese Academy of Sciences | XDA19050101 XDB26000000 | Bo Wang |
| National Natural Science Foundation of China | 42125201 41688103 | Bo Wang |
| Natural Scientific Foundation of Shandong Province | ZR2020YQ27 | Jun Chen |
| Russian Science Foundation | 21-14-00284 | Evgeny Yan |
| Chinese Academy of Sciences | 2020VCA0020 | Edmund A Jarzembowski |

The funders had no role in study design, data collection and interpretation, or the decision to submit the work for publication.

### Author contributions

Xianye Zhao, Data curation, Formal analysis, Investigation, Methodology, Software, Validation, Visualization, Writing - original draft, Writing - review and editing; Yilun Yu, Formal analysis, Investigation, Methodology, Software, Validation, Visualization, Writing - original draft, Writing - review and editing; Matthew E Clapham, Investigation, Validation, Writing - review and editing; Evgeny Yan, Investigation, Visualization, Writing - review and editing; Jun Chen, Formal analysis, Investigation, Methodology, Writing - review and editing; Edmund A Jarzembowski, Xiangdong Zhao, Investigation, Writing - review and editing; Bo Wang, Conceptualization, Data curation, Formal analysis, Funding acquisition, Investigation, Methodology, Project administration, Resources, Supervision, Validation, Visualization, Writing - original draft, Writing - review and editing

## Author ORCIDs

Bo Wang (ID) http://orcid.org/0000-0002-8001-9937

## Decision letter and Author response

Decision letter https://doi.org/10.7554/eLife.72692.sa1
Author response https://doi.org/10.7554/eLife.72692.sa2

## Additional files

### Supplementary files
- Supplementary file 1. Taxonomic revisions.
- Supplementary file 2. List of taxa used for the phylogenetic analysis.
- Supplementary file 3. Characters used for the phylogenetic analysis.
- Supplementary file 4. Characters used for the morphospace analysis.
- Transparent reporting form

### Data availability

All source data are available at https://doi.org/10.5061/dryad.7m0cfxpvd. In addition, the source data files (Supplementary Data 1-4) have been provided for figures 2-4 and appendix figures 1-10.

The following dataset was generated:

| Author(s) | Year | Dataset title | Dataset URL | Database and Identifier |
|---|---|---|---|---|
| Zhao X, Yu Y, Clapham M, Yan E, Chen J, Jarzembowski E, Zhao X, Wang B | 2021 | Dataset of Early evolution of beetles regulated by the end-Permian deforestation | https://doi.org/10.5061/dryad.7m0cfxpvd | Dryad Digital Repository, 10.5061/dryad.7m0cfxpvd |

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
