## [Editor Report]

The study proposes a new evolutionary-ecological scenario for Late Paleozoic and early Mesozoic beetles, supported by the summary of all available knowledge about early beetle fossils, including analyses of their taxon and morphological diversity and phylogenetic relationships. The effects of xylophagous beetles during the Paleozoic may have played a fundamental role in global biochemical cycles. The results advance our understanding of the evolutionary success of beetles and the many ways in which large environmental changes may affect biodiversity in general.

---

## [Decision Letter]

**Decision letter after peer review:**

Thank you for submitting your article "Early evolution of beetles regulated by the end-Permian deforestation" for consideration by *eLife*. Your article has been reviewed by 2 peer reviewers, and the evaluation has been overseen by George Perry as the Senior and Reviewing Editor. The following individual involved in review of your submission has agreed to reveal their identity: Martin Fikacek (Reviewer #3).

The reviewers have discussed their reviews with one another and the Reviewing Editor. The two reviewers have provided clear and excellent feedback regarding essential and suggested changes in their individual reviews; I refer you to their comments below to help you prepare a revised submission to *eLife*.

*Reviewer #2 (Recommendations for the authors):*

Lines:

31 remove "a" "… by xylophagous stem groups with high diversity and disparity, ".

182 McKenna, not Mckenna.

207 change to "consumed".

208-211 Sentence from lines 208-211 could be rephrased to:

Some Permian xylophagous beetles fed on living wood tissues (Feng et al., 2017, 2019), which likely reduced tree productivity and could have caused extensive tree mortality. Insect-mediated tree mortality is known to result in large transfers of carbon from biomass to dead organic matter (Seidl et al., 2018; Fei et al., 2019).

The original text made it sound like there was data on this carbon transfer in the Permian.

215 add commas to "small invertebrates, such as oribatid mites, until the".

243-245 This is pretty correlative. A stronger argument would look at the other proposed causes of decreasing O2 in the Early Permian (ex. fires, see Glasspool 2006) and show a stronger relationship between atm O2 and the evolution of beetle xylophagy.

The diversification of Paleozoic fire systems and fluctuations in atmospheric oxygen concentration. August 2006 Proceedings of the National Academy of Sciences 103(29):10861-5.

246 change to something like "…been responsible for at least a portion of the change of biogeochemical cycles of carbon…".

253-255 Cite the statement for “Permian beetles were the principal degraders of wood”, and/or say something like “… among the principal”..".

280 include "and" between the Vajada et al., 2020 citation and "were".

*Reviewer #3 (Recommendations for the authors):*

This is a very nicely done and written study, I definitely like the multi-approach and multi-evidence approach which together brings a very clear view and very well-supported hypotheses about the early evolution of beetles and their roles in the ancient ecosystems. I provided my "praise" already in the public review, so here I will focus on some critique and recommendations.

My specialization is beetle systematics and phylogenetics, so the part I can comment on most in detail is naturally the phylogenetic analysis. As I mentioned in the public statement, I agree that the analysis provided is good enough for this study, which was not focused only on the phylogeny and which aims at the more general view. However, I did not find any information on whether the actual species included in the analysis were really reexamined and whether the characters included were double-checked (which would be the ideal way), or whether the dataset was simply adopted from Beutel (2008) and updated by few new characters and few new taxa (which is the way most people do, but it naturally risks that some mistakes of misinterpretations present in the published matrices are adopted and copied again and again). As far as I knew, Beutel´s (2008) dataset was compiled largely from the literature without reexamining the actual fossils, which is why it should be used with care.

If I can ask about one specific character: you follow the view presented in old papers (and adopted in the matrix by Beutel 2008) that the earliest beetles had 13-segmented antennae. This is really illustrated in many reconstructions by Ponomarenko, but as far as I remember, most of these fossils actually do not have antennae preserved. There were I think only two fossils of Tschekardocoleidae which antennae were preserved, and from the photos available in the literature, I had strong doubts about their antennae being 13-segmented, it always looked like normal 11-segmented beetle antennae. I am asking not only because this character is included in the morphology matrix (as a single character, it would likely not change the topology too much even when coded incorrectly), but also because you provide the new reconstruction of the Tschekardocoleid beetles (again with 13-segmented antennae), so it would be good to double-check this character to be sure you are just not reproducing the old mistake again and again.

In general, I also believe that including more taxa into the analysis would be desirable, not just a single taxon per family, but I agree that for this study you need a simplified phylogeny only and your approach hence can be considered as justified.

Few other questions:

(1) You state that Tschekardocoleidae is the oldest beetle, but you do not mention Coleopsis archaica anywhere in the paper (sorry if I overlooked it). Does it mean you do not consider this fossil as the representative of Coleoptera? It would be good to provide some sentence about that, to make it clear for the reader why this fossil is not considered (e.g., if you directly exclude it from beetles, or if you just consider it as doubtful and hence better do not include it in your analyses).

(2) There are also larval characters in your morphology matrix, but it is unclear to me which larvae were coded - can you provide some information in your list of taxa included in the analysis about which taxon has only adult data, and which has also larval data? And in the case of larvae coded, can you provide some justification of why the larva you coded is supposed to belong to the respective adult? In fossils, associating larvae with adults is a tricky task and often the associations are kind of guesses, so it would be good to know your arguments supporting the larva-adult associations for the taxa in which you coded larvae.

(3) You compare fossils data about the origin of some beetles groups with the molecular time trees results - that is perfect. But better cite more than one time tree study (you only mention McKenna 2019). This is because all molecular time trees are subject to bias and mistakes, partly introduced by specific methods used, partly by more or less carefully done selection of calibrating fossils and the way how they are implemented in the study. All that means, that we can be confident basically only about the results which are revealed repeatedly in various independent timetree studies, and hence ideally a reference to multiple studies is needed.

(4) I am not a specialist in beetle ecology and environmental sciences, but when reading the discussion connecting the early beetles to carbon cycling and dead-wood decay, I got some questions for which I did not find any reply in the text:

– You state in the introduction that since all early beetle lineages are extinct, it is hard to apply the knowledge about modern beetle lineages fully - I totally agree with this, the phylogenetic position of the early beetles is so basal that the simple parallels with modern beetles do not need to work. But after (correctly) stating that in the introduction, you use precisely the "parallel with modern beetles" approach to state that the ancient beetles were wood-boring because they looked like modern wood-boring Archostemata. I kind of feel that there are two problems hidden in this: (1) the parallel "looks same so it lived the same" may not necessarily work for 100% (as you stated in the introduction), and (2) I am not sure how much is actually known about the detailed biology of modern Archostemata (I know they are referred as wood-boring in all general chapters but are there actually some studies about their lifestyle?). I am not saying that these issues disqualify your conclusions, but maybe a more careful phrasing, and in some cases, more detailed and direct support of some statements (like wood-boring Archostemata) would make the text easier to read and would present your conclusions as more sound.

– I am not really good at general paleontology and geology etc., so reading your general discussions about environmental changes at P-T was interesting. But I would maybe sometimes welcome to have a little bit more information provided, not to just found the reference to the paper. Some parts are really cool lists and summaries of recently published studies, but it would be simply beneficial for readers without general paleo-knowledge like me not to need to dig into all these papers to understand your conclusions. I know there likely is a strong length limitation for the manuscript, but I ask even the editor, in this case, to make it possible to provide a little bit more information than just a list of references.

---

## [Author Response]

Reviewer #2 (Recommendations for the authors):Lines:31 remove "a" "… by xylophagous stem groups with high diversity and disparity, ".

Thank you. We have corrected it. Please see line 31.

182 McKenna, not Mckenna.

Thank you. We have corrected it. Please see line 187.

207 change to "consumed".

Thank you. We have corrected it. Please see line 214.

208-211 Sentence from lines 208-211 could be rephrased to:Some Permian xylophagous beetles fed on living wood tissues (Feng et al., 2017, 2019), which likely reduced tree productivity and could have caused extensive tree mortality. Insect-mediated tree mortality is known to result in large transfers of carbon from biomass to dead organic matter (Seidl et al., 2018; Fei et al., 2019).The original text made it sound like there was data on this carbon transfer in the Permian.

Thank you and revised. Please see lines 214–218. “Some Permian xylophagous beetles fed on living wood tissues (*Feng et al., 2017, 2019*), which likely reduced tree productivity and could have caused extensive tree mortality. Insect-mediated tree mortality is known to result in large transfers of carbon from biomass to dead organic matter (*Seidl et al., 2018*; *Fei et al., 2019*).”

215 add commas to "small invertebrates, such as oribatid mites, until the".

Thank you. We have corrected it. Please see line 223.

243-245 This is pretty correlative. A stronger argument would look at the other proposed causes of decreasing O2 in the Early Permian (ex. fires, see Glasspool 2006) and show a stronger relationship between atm O2 and the evolution of beetle xylophagy.The diversification of Paleozoic fire systems and fluctuations in atmospheric oxygen concentration. August 2006 Proceedings of the National Academy of Sciences 103(29):10861-5.

Thank you. Scott and Glasspool (2006) concluded that oxygen levels are a significant control on long-term fire occurrence and suggested that more fire may lead to increased charcoal production and further increased levels of oxygen. However, they did not refer to the reason about the decrease of oxygen. So we did not cite this reference. We have added other two hypotheses about the reason for the oxygen plunge (reduction in the extent of coal swamps and the evolution of lignin-consuming fungi) and ruled them out in the Discussion. Please see comment 1.

246 change to something like "…been responsible for at least a portion of the change of biogeochemical cycles of carbon…".

Thank you and revised. Please see line 272–273.

253-255 Cite the statement for “Permian beetles were the principal degraders of wood”, and/or say something like “… among the principal”..".

Thank you. We have add “… *among* the principal degraders…”. Please see line 280.

280 include "and" between the Vajada et al., 2020 citation and "were".

Thank you. We have corrected it. Please see line 307.

Reviewer #3 (Recommendations for the authors):This is a very nicely done and written study, I definitely like the multi-approach and multi-evidence approach which together brings a very clear view and very well-supported hypotheses about the early evolution of beetles and their roles in the ancient ecosystems. I provided my "praise" already in the public review, so here I will focus on some critique and recommendations.My specialization is beetle systematics and phylogenetics, so the part I can comment on most in detail is naturally the phylogenetic analysis. As I mentioned in the public statement, I agree that the analysis provided is good enough for this study, which was not focused only on the phylogeny and which aims at the more general view. However, I did not find any information on whether the actual species included in the analysis were really reexamined and whether the characters included were double-checked (which would be the ideal way), or whether the dataset was simply adopted from Beutel (2008) and updated by few new characters and few new taxa (which is the way most people do, but it naturally risks that some mistakes of misinterpretations present in the published matrices are adopted and copied again and again). As far as I knew, Beutel´s (2008) dataset was compiled largely from the literature without reexamining the actual fossils, which is why it should be used with care.If I can ask about one specific character: you follow the view presented in old papers (and adopted in the matrix by Beutel 2008) that the earliest beetles had 13-segmented antennae. This is really illustrated in many reconstructions by Ponomarenko, but as far as I remember, most of these fossils actually do not have antennae preserved. There were I think only two fossils of Tschekardocoleidae which antennae were preserved, and from the photos available in the literature, I had strong doubts about their antennae being 13-segmented, it always looked like normal 11-segmented beetle antennae. I am asking not only because this character is included in the morphology matrix (as a single character, it would likely not change the topology too much even when coded incorrectly), but also because you provide the new reconstruction of the Tschekardocoleid beetles (again with 13-segmented antennae), so it would be good to double-check this character to be sure you are just not reproducing the old mistake again and again.In general, I also believe that including more taxa into the analysis would be desirable, not just a single taxon per family, but I agree that for this study you need a simplified phylogeny only and your approach hence can be considered as justified.

Thank you. We completely agreed with the reviewer that most of early beetles need re-examination. We examined some early beetle fossils deposited in Moscow, and found that the original descriptions of some species are not correct probably due to the poor preservation. Therefore, we only use the best-preserved species to represent the higher-level taxa (family). A comprehensive revision of all fossils are very important, but is beyond the scope of our paper. Regarding the phylogeny, we mainly followed the Beutel’s (2008) dataset, which is the only dataset for the fossil beetles. As the statement of the reviewer, the simplified phylogeny is enough for illustrating the effect of end-Permian extinction to the beetle evolution.

We are grateful to the reviewer to point out the issue about the segmentation of the antennae. We re-examined the Russian specimens and discussed this issue with several colleagues including Rolf Beutel. We completely agreed with the reviewer that Tschekardocoleidae and Permocupedidae have most likely only 11-segmented antennae. We have revised the matrix and figures (Fig. 1B, D). We also carefully re-examined the matrix of the phylogenetic analysis and corrected several characters. Our new phylogenetic result is almost the same as the previous one (Figure 3-figure supplement 1).

Few other questions:(1) You state that Tschekardocoleidae is the oldest beetle, but you do not mention Coleopsis archaica anywhere in the paper (sorry if I overlooked it). Does it mean you do not consider this fossil as the representative of Coleoptera? It would be good to provide some sentence about that, to make it clear for the reader why this fossil is not considered (e.g., if you directly exclude it from beetles, or if you just consider it as doubtful and hence better do not include it in your analyses).

Thank you very much for pointing out this issue. *Coleopsis archaica* was included in our database. This species was attributed to Tschekardocoleidae by Kirejtshuk et al. (2014), but later was elevated to a new family Coleopseidae by Kirejtshuk (2020). We followed the former opinion because it is premature to erect a family without a detailed cladistic analysis. This has been clarified in the revised version of the manuscript. Please see line 184. “The earliest definite beetles are Tshekardocoleidae (including the genus *Coleopsis*) from the Early Permian …”. Please see lines 353–356. “The species *Coleopsis archaica* was attributed to Tschekardocoleidae by Kirejtshuk et al. (2014), but was later elevated to a new family Coleopseidae by Kirejtshuk (2020). We followed the former opinion because it is premature to erect a family without a detailed cladistic analysis.”

We also added a reference.

Kirejtshuk AG. 2020. Taxonomic review of fossil coleopterous families (Insecta, Coleoptera). Suborder Archostemata: superfamilies Coleopseoidea and Cupedoidea. Geosciences 10: 73.

(2) There are also larval characters in your morphology matrix, but it is unclear to me which larvae were coded - can you provide some information in your list of taxa included in the analysis about which taxon has only adult data, and which has also larval data? And in the case of larvae coded, can you provide some justification of why the larva you coded is supposed to belong to the respective adult? In fossils, associating larvae with adults is a tricky task and often the associations are kind of guesses, so it would be good to know your arguments supporting the larva-adult associations for the taxa in which you coded larvae.

Thank you. In the phylogeny dataset, larval characters of all extinct groups were coded “unknown”, due to the lack of larval fossils.

(3) You compare fossils data about the origin of some beetles groups with the molecular time trees results - that is perfect. But better cite more than one time tree study (you only mention McKenna 2019). This is because all molecular time trees are subject to bias and mistakes, partly introduced by specific methods used, partly by more or less carefully done selection of calibrating fossils and the way how they are implemented in the study. All that means, that we can be confident basically only about the results which are revealed repeatedly in various independent timetree studies, and hence ideally a reference to multiple studies is needed.

We are grateful for the suggestion. We have added two references to the Discussion section.

(4) I am not a specialist in beetle ecology and environmental sciences, but when reading the discussion connecting the early beetles to carbon cycling and dead-wood decay, I got some questions for which I did not find any reply in the text:– You state in the introduction that since all early beetle lineages are extinct, it is hard to apply the knowledge about modern beetle lineages fully - I totally agree with this, the phylogenetic position of the early beetles is so basal that the simple parallels with modern beetles do not need to work. But after (correctly) stating that in the introduction, you use precisely the "parallel with modern beetles" approach to state that the ancient beetles were wood-boring because they looked like modern wood-boring Archostemata. I kind of feel that there are two problems hidden in this: (1) the parallel "looks same so it lived the same" may not necessarily work for 100% (as you stated in the introduction), and (2) I am not sure how much is actually known about the detailed biology of modern Archostemata (I know they are referred as wood-boring in all general chapters but are there actually some studies about their lifestyle?). I am not saying that these issues disqualify your conclusions, but maybe a more careful phrasing, and in some cases, more detailed and direct support of some statements (like wood-boring Archostemata) would make the text easier to read and would present your conclusions as more sound.

Thank you very much. We completely agreed with the reviewer that we are not 100% certain that these extinct beetles are wood-boring. We deduced their palaeoecology based on morphological comparisons and potential trace fossils, following the previous studies (e.g., *Ponomarenko, 1969*; *Naugolnykh and Ponomarenko*, 2010). We toned down the statement of the palaeoecology throughout the Discussion part. Please see lines 200 and 213. “These ancient beetles were *most likely* xylophagous”. “most Permian beetles were *most likely* xylophagous insects”.

– I am not really good at general paleontology and geology etc., so reading your general discussions about environmental changes at P-T was interesting. But I would maybe sometimes welcome to have a little bit more information provided, not to just found the reference to the paper. Some parts are really cool lists and summaries of recently published studies, but it would be simply beneficial for readers without general paleo-knowledge like me not to need to dig into all these papers to understand your conclusions. I know there likely is a strong length limitation for the manuscript, but I ask even the editor, in this case, to make it possible to provide a little bit more information than just a list of references.

Thank you. There are two paragraphs about the P-T environmental changes in the Discussion section. We followed the reviewer’s suggestion and enlarged both paragraphs.

Regarding the Permian environment, we have added an introduction about the change of atmospheric oxygen and included several hypothesis. Please see lines 237–249. “The oxygen concentration of the atmosphere began to rise in the early Palaeozoic, probably with a peak in the Carboniferous and large decline from the beginning of the Permian (Dahl et al., 2010; Berner, 2009; Krause et al, 2018). The reason for this plunge was attributed to a tectonic- or climatic-forced reduction in the extent of coal swamps (Berner and Canfield, 1989) or to the evolution of lignin-consuming fungi (Floudas et al., 2012). However, global recoverable coal is only equivalent to a few percent of the oxygen budget in the atmosphere, and thus cannot account for the large drop of atmospheric oxygen (Nelsen et al., 2016). Furthermore, lignin-consuming fungi may have been present before the Carboniferous (Nelsen et al., 2016). Recently, a new geochemical model proposed that the development of Permian terrestrial herbivores may have limited transport and long-term burial of terrestrial organic compounds in marine sediments, resulting in less organic carbon burial and attendant declines in atmospheric oxygen (Laakso et al., 2020).”

Regarding the Early Triassic coal gap, we have added a brief introduction. Please see lines 302–306. “This gap in xylophagous beetles coincided chronologically with the gap in coal deposition (“coal gap”), a time during which peat-forming forests were rare or absent (Figure 3B), extending across at least the entire Early Triassic (Benton and Newell, 2014, Retallack et al., 1996; Nowak et al., 2020; Zhao et al., 2020).”